# Torch$T_1$: GPU-accelerated cardiac $T_1$ mapping with deep learning framework

**Yi Zhang**[1]                                                     Y.ZHANG-43@TUDELFT.NL
**Yidong Zhao**[1]                                                    Y.ZHAO-8@TUDELFT.NL
**Yifeng Shao**[1]                                                       Y.SHAO@TUDELFT.NL
**Nuno Miguel Ferreira Capitão**[1]                       NNUNOMIGUELFER@TUDELFT.NL
**Fleur van den Bogert**[1]                     F.VANDENBOGERT@STUDENT.TUDELFT.NL
**Qian Tao**[1*]                                                            Q.TAO@TUDELFT.NL

[1] *Department of Imaging Physics, Delft University of Technology, The Netherlands*

## Abstract

Quantitative cardiac $T_1$ mapping by MRI is an essential non-invasive diagnostic tool for cardiomyopathies. Traditionally, deriving the quantitative $T_1$ maps of myocardial tissue involves solving non-linear parametric fitting problems per image voxel, which is slow with sequential CPU computation and requires analytical derivation of the Jacobian matrix per signal model. In this paper, we introduce a new paradigm of parametric fitting, termed "Torch$T_1$", which leverages the powerful parallelization of modern GPUs and well-established functionalities of auto-differentiation in the deep learning framework of PyTorch. Torch$T_1$ strictly adheres to the signal model and does not require any training. Our method was evaluated on a $T_1$ mapping dataset with both pre-contrast and post-contrast sequences, and benchmarked by conventional CPU-based fitting and recent end-to-end physics-informed neural network (PINN) mapping. Torch$T_1$ showed more accurate and reliable mapping quality compared with the pretrained PINN, with a 13-fold acceleration compared with the CPU baseline.

**Keywords:** Quantitative $T1$ mapping, MRI, Automatic differentiation, PyTorch.

## 1. Introduction

Quantitative cardiac MRI, including $T_1$ and $T_2$ mapping (Messroghli et al., 2004; O'Brien et al., 2022), is increasingly important in non-invasive diagnosis of cardiomyopathies (Haaf et al., 2016). The quantitative maps are derived by fitting a parametric model to a sequence of baseline images acquired under specific MR protocols, often called *mapping*. Parametric mapping is an optimization problem, traditionally addressed either through simplex search methods like the Nelder-Mead algorithm (Nelder and Mead, 1965) or gradient-based methods like the Levenberg-Marquardt algorithm (Gavin, 2019). Typically, MRI mapping is performed in a serial manner, voxel by voxel across the field of view, with each voxel involving an iterative optimization. Mapping the whole imaging field thus becomes quadratically slow when the image size increases. Acceleration is possible with parallel CPUs but limited by the number of cores.

---

* Corresponding author

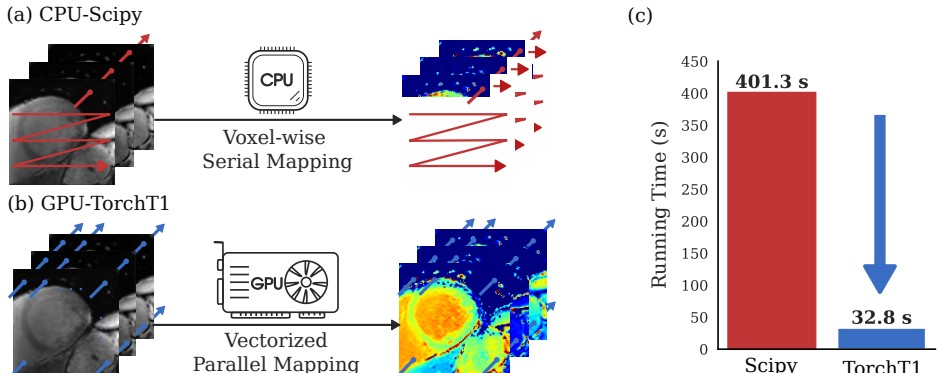

Figure 1: Comparison of the two methodologies: (a) CPU-based Scipy employs serial computing; (b) GPU-based Torch$T_1$ performs parallel computing. The average computation time for one $T_1$ sequence is reported in (c) for Scipy and Torch$T_1$.

We propose a new paradigm of quantitative mapping, termed "Torch$T_1$", which leverages the powerful parallelization of modern GPUs, and well-established functionalities of auto-differentiation (AD) (Paszke et al., 2019) and gradient descent (GD) (Kingma and Ba, 2014) in the state-of-the-art deep learning framework. Different from recent deep learning work that learns the mapping from data (Guo et al., 2022) or incorporates physics priors (Sabidussi et al., 2021), Torch$T_1$ strictly adheres to the physics model and does not require any training. It solves the optimization problem with all voxels in parallel, leveraging the powerful AD and GD modules of PyTorch (Paszke et al., 2019) without the need to analytically derive the Jabcobian. This makes our method well scalable in two senses: first, to larger image size; second, to new physics models of quantitative mapping with an arbitrary number of parameters (Chow et al., 2022; Božić-Iven et al., 2024). Figure 1 shows a comparison between the conventional CPU mapping and our proposed Torch$T_1$ mapping.

## 2. Method and Experiments

In a mapping sequence, $N$ baseline images are acquired, each at a different acquisition setting. We denote $S_{i,j}$ to be the measured signal at voxel $i \in \{1, 2, \ldots, M\}$ for $j$-th baseline image where $j \in \{1, 2, \ldots, N\}$. We consider $T_1$ mapping with the widely adopted Modified Look-Locker inversion recovery (MOLLI) sequence (Messroghli et al., 2004), with a 3-parameter signal model:

$$S_{i,j} = \left| C_i \left( 1 - k_i \exp\left( -\frac{t_j}{T_{1\,i}^*} \right) \right) \right|, \tag{1}$$

where $t_j$ is the inversion time of $j$-th image, the parameter set $\left\{ C_i, k_i, T_{1\,i}^* \right\}$ are mapped at voxel $i$ to derive the tissue property $T_{1i} = (k_i - 1)T_{1\,i}^*$. With the proposed Torch$T_1$, we estimate the three parameters by minimizing the mean square error (MSE) between estimated signal $\widehat{S}_{i,j}$ and true signal $S_{i,j}$, through gradient descent. For gradient calculation, we use the AD functionality in PyTorch to calculate the Jacobian of Eq.1. For parallel

computation, we treat the three parameters as independent vectors $C = [C_1, C_2, \ldots, C_M]^\top$, $k = [k_1, k_2, \ldots, k_M]^\top$, and $T_1^* = [T_1^*{}_1, T_1^*{}_2, \ldots, T_1^*{}_M]^\top$. Each entry of the parameter vector can be processed in parallel (*i.e.*, a calculation taking $[C_1, k_1, T_1^*{}_1]$ as input and their update as output), thanks to the well-established GD functionalities in the deep learning framework, allowing the estimated signals at time $t_j$, $\widehat{S}_j = [\widehat{S}_{1,j}, \widehat{S}_{2,j}, \ldots, \widehat{S}_{M,j}]^\top$ to be computed in parallel. This implies that the mapping of all voxels in the field of view is done simultaneously, instead of serially, as illustrated in Fig. 1.

We evaluated the proposed TorchT₁ on a cardiac MRI dataset with 30 pre- and 30 post-contrast MOLLI sequences (Philips 3.0T). We used the ADAM optimizer with an initial learning rate of $3 \times 10^{-4}$. We compared our method with a bounded Nelder-Mead algorithm with the same MSE loss in Scipy (Virtanen et al., 2020) as a baseline. In addition, we compared the results with a pretrained physics-informed neural network (PINN) (Guo et al., 2022). All experiments were run on the same workstation (Intel Xeon 3.9GHz 8 threads, NVIDIA RTX 4090, 80GB RAM).

## 3. Results and Conclusions

We compare the $T_1$ mapping accuracy of the baseline Scipy method and our TorchT₁ in Fig. 2 (a). TorchT₁ demonstrated consistent accuracy in both pre-contrast and post-contrast mapping, with significantly reduced computation time from 401.3s to 32.8s as shown in Fig. 1 (b). Fig. 2 (b) further shows TorchT₁'s visual quality compared to the Scipy baseline. Unlike the pretrained PINN in Fig. 2 (b) which shows strong bias, TorchT₁ operates without training, ensuring reliability across varied acquisition settings.

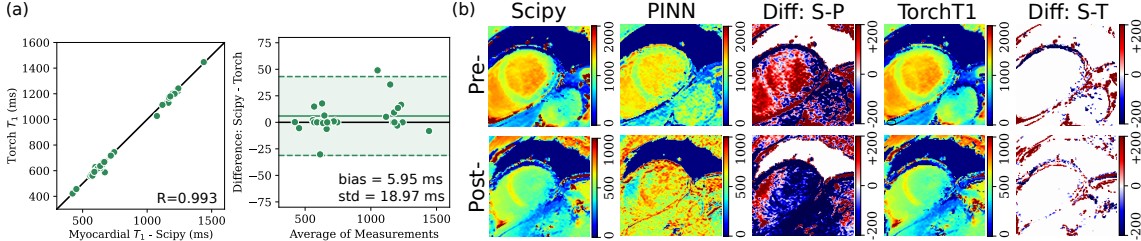

Figure 2: (a) Bland-Altman plot of the average myocardial $T_1$ values by TorchT₁ and Scipy. (b) $T_1$ values estimated by Scipy, PINN, and TorchT₁, and their differences.

In conclusion, the accuracy and reliability of the proposed GPU-based TorchT₁ are highlighted by our preliminary quantitative results compared with the conventional implementation on CPU, with substantial speed acceleration. The qualitative results also affirm the acclaimed reliability against the pretrained end-to-end PINN which can be biased due to potential domain shifts across signal models and acquisition settings, suggesting that our GPU-based TorchT₁ can serve as a fast and reliable framework for cardiac $T_1$ mapping. TorchT₁ can potentially be extended to other quantitative MRI sequences given its generic formulation.

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
