# OpenReview forum: "Torch$T_1$: GPU-accelerated cardiac $T_1$ mapping with deep learning framework"
_MIDL.io/2024/Short_Papers — MIDL 2024 Short Papers_

### Official Review · Reviewer_4gEN · 2024-04-24

**Confidence:** 4
**Final Rating:** 4

**Review:**

This paper introduces a new method, termed TorchT1, that uses the efficiency of GPU parallelization and PyTorch auto-differentiation to accelerate the computation of voxelwise parametric fitting problems for T1 mapping.

Pros:

The paper is well-written and easy for readers to follow. The goal of accelerating T1 mapping with high accuracy is appealing and interesting.

Cons:

The experimental evaluation lacks clarity, particularly in the comparison to a pre-trained PINN. It might be more appropriate to compare with a fine-tuned PINN on the authors' dataset to ensure a fair comparison of performance.

---

### Decision · Program_Chairs · 2024-04-26

Accept